# Client Security Alone Fails in Federated Learning: 2D and 3D Attack Insights

Anoymous Author

`author@org.com`

**Abstract.** Federated learning (FL) plays a vital role in boosting both accuracy and privacy in the collaborative medical imaging field. The importance of privacy increases with the diverse security standards across nations and corporations, particularly in healthcare and global FL initiatives. Current research on privacy attacks in federated medical imaging focuses on sophisticated gradient inversion attacks that can reconstruct images from FL communications. These methods demonstrate potential worst-case scenarios, highlighting the need for effective security measures and the adoption of comprehensive zero-trust security frameworks. Our paper introduces a novel method for performing precise reconstruction attacks on the private data of participating clients in FL settings using a malicious server. We conducted experiments on brain tumor MRI and chest CT data sets, implementing existing 2D and novel 3D reconstruction technique. Our results reveal significant privacy breaches: 35.19% of data reconstructed with 6 clients, 37.21% with 12 clients in 2D, and 62.92% in 3D with 12 clients. This underscores the urgent need for enhanced privacy protections in FL systems. To address these issues, we suggest effective measures to counteract such vulnerabilities by securing gradient, analytic, and linear layers. Our contributions aim to strengthen the security framework of FL in medical imaging, promoting the safe advancement of collaborative healthcare research. The source code is available at: `https://www.github.com/anonymous`.

**Keywords:** Federated Learning · Privacy Attacks · Medical Imaging · Reconstruction Attacks

## 1 Introduction

Privacy and regulatory challenges limit the gathering of large medical datasets for deep neural network training [11]. Federated Learning (FL) [12] addresses these challenges by enabling collaborative training across hospitals, without centralizing patient data. Hospitals train models locally, share only model updates with a central server, which then aggregates these to improve a global model. Although patient data remains distributed in FL, there is still a significant risk of data leakage through information encoded in model updates, which may be exploited by attackers to reconstruct training data and hence poses a major privacy concern (e.g. in the case of images by facial reconstruction from MRI data [18]).

With the rise in popularity of FL, its associated software and frameworks, including NVFlare[15], Kaapana[10], MonaiFL[13], and Flower[1], have also gained prominence. Alongside this growth, various privacy attacks have been proposed. This paper focuses on data reconstruction attacks, which aim to recover private training data points as accurately as possible. These attacks utilize methods such as optimization [7,25,23,6,4], analytical techniques [22,16], or exploit vulnerabilities in linear layers [2,5] to compromise FL privacy. Medical FL research has mainly focused on threats through gradient-based optimization [24,11,21,8] and model inversion [20] techniques. These strategies face major challenges, such as high computational resource demands, and typically result in incomplete or blurred reconstructions. This low reconstruction fidelity poses challenges in accurately identifying original records despite being indicative of a privacy breach and is especially relevant in medical imaging, where subtle structural variances are often crucial.

Our research investigates the potential of data reconstruction attacks in FL that can closely replicate the private data of patients, in the scenario of a malicious FL server. We apply an analytical approach[2] to 2D medical imaging and introduce a novel adaptation for 3D medical imaging within the context of FL. This marks the first effort in 3D imaging, achieving superior reconstruction fidelity with minimal architecture modifications from a malicious server. Our study underscores the need for measures to prevent data leakage through malicious server activities, even in the context of a secure client, in terms of software and network aspects. The used attack highlights the importance of deeply understanding the machine learning architecture involved, when an institution decides to participate in a federated learning collaboration. In summary, our study presents the subsequent contributions:

- To our knowledge, this is the first demonstration of privacy attacks in FL using 3D medical imaging data with malicious server-shared gradients. Our method reconstructs slices into 3D volumes with minimal modifications, highlighting significant privacy concerns. In simulations with 6 and 12 clients, we achieve high-fidelity 2D and 3D reconstructions without complex optimization, showing the potential for reconstructing identifiable body parts or whole bodies.[18].
- We showcase that irrespective of client security, a malicious server can easily attack clients as long as the server or central entity provides the training algorithm, which is common in most FL scenarios.
- We also outline simple but effective techniques to mitigate malicious server-based attacks in cross-consortia medical FL.

## 2   Method

In Federated Learning with a central server, the server can introduce vulnerabilities when distributing models to clients. This study adapts a 2D attack method from [2] to 3D, where a malicious server inserts a linear layer with ReLU activa-

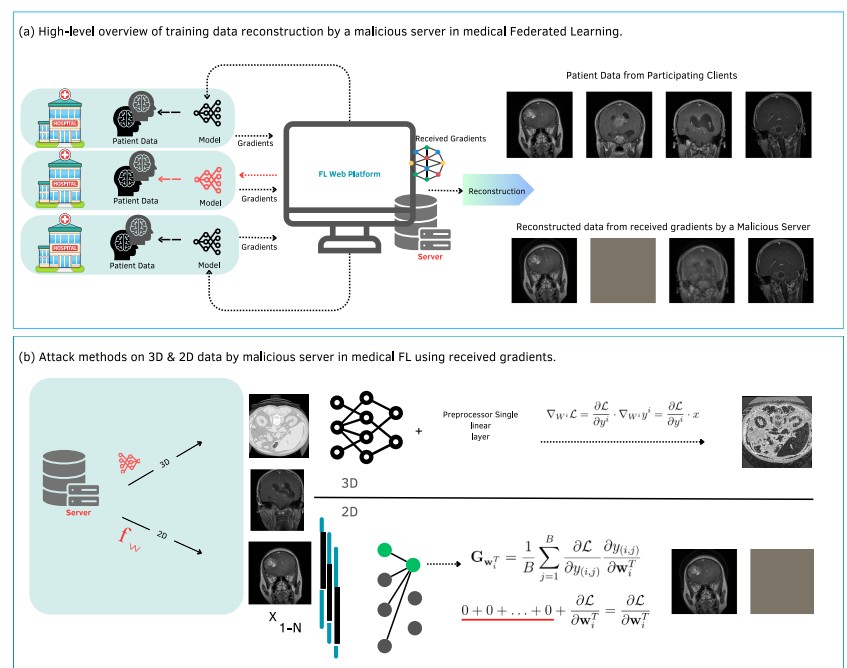

**Fig. 1.** (a) Illustrates the federated medical imaging setup with hospitals connecting to a central server via a web platform for algorithm exchange and gradient sharing, mimicking real-world practices. (b) Depicts the central server's attack on client 3D and 2D medical imaging data through model preprocessing and malicious weight adjustments, demonstrating successful reconstruction of both data types.

tion before the first original network layer to extract sensitive image information from the clients' model updates.

**Data Reconstruction Overview:** Let $y_i = w_i^T x + b_i$, the output of neuron $i$ in the inserted linear layer. Then the input $x$ can be reconstructed from the gradients of the loss $L$ with respect to both the bias and the weights in the following manner[6,2]:

$$\frac{\partial L}{\partial b_i} = \frac{\partial L}{\partial y_i}\frac{\partial y_i}{\partial b_i} = \frac{\partial L}{\partial y_i}, \quad \frac{\partial y_i}{\partial b_i} = 1, \tag{1}$$

$$\frac{\partial L}{\partial w_i^T} = \frac{\partial L}{\partial y_i}\frac{\partial y_i}{\partial w_i^T} = \frac{\partial L}{\partial b_i}x^T, \tag{2}$$

assuming only a single sample $x$ is passed through the network. If any $\frac{\partial L}{\partial b_i} \neq 0$, perfect reconstruction is achieved by:

$$x^T = \left(\frac{\partial L}{\partial b_i}\right)^{-1}\frac{\partial L}{\partial w_i^T} \tag{3}$$

where ReLU activation is $> 0$. This shows that the input data can be retrieved from the gradients of the linear layer at the front which receives that data as input. When using mini batches, for the $i^{\text{th}}$ neuron, the batch gradient is represented by:

$$G_{w_i}^T = \frac{1}{B} \sum_{j=1}^{B} \frac{\partial L}{\partial y_{(i,j)}} \frac{\partial y_{(i,j)}}{\partial w_i^T} \tag{4}$$

Under the condition that all but one term in equation (4) vanish, individual datapoints can be reconstructed from minibatch gradients. As [2] point out, this happens when the ReLU activations are zero for all but one input, allowing to reconstruct samples even when $B > 1$. They further show that reconstruction is even possible when using FedAvg. Initializing the layer with malicious weights allows to increase the chances for the exact reconstruction of individual datapoints. In our experiments, A malicious server added an attack layer, which effectively inserted a linear layer before the actual network. A single federated optimization step was simulated, after which gradients were shared with the server. The server used the gradients from the attack layer to invert the fully connected layer's gradients for data reconstruction.

**Adaptation to 3D Data:** We introduce a methodology within the federated learning round where a malicious server employs a custom preprocessor to execute attacks. This preprocessor converts the original 3D volume into 2D slices, facilitating the server's malicious algorithm. This approach extends the 2D attack by injecting specialized preprocessing code, enabling spatial mapping of the extracted data to the corresponding slices to accurately reconstruct the 3D volume. The server then applies the same attack method described in equation (3) to reconstruct the 2D slices. These slices are then reassembled into one 3D volume of the target client's data, as we are focusing on a single client's single 3D volume data during a specific epoch. By incorporating trap weights into the initial fully connected (FC) layer, we can increase the attack's accuracy when targeting a specific client's specific batch. For weight row $w_i$, with $N$ and $P$ indicating negative and positive weights, a neuron activates with ReLU if negative-weighted sums are less than positive-weighted sums. Due to this malicious weight initialisation 5 hold rare inputs, typically only affecting a single data point per mini-batch which can be then extracted using 3.

$$\sum_{n \in N} w_{n,i} x_n < \sum_{p \in P} w_{p,i} x_p. \tag{5}$$

## 3  Experiments

**Brain Tumor MRI Dataset (2D Attack):** [3] contains 7023 MRI brain tumor slices with dimensions $512 \times 512$. Each slice is labeled with one of the 4 classes Glioma, Meningioma, Pituitary, and No Tumor. For the 6-client experiment, we used a total of 1800 images (300 per client). For the 12-client experiment, we used 2640 images (220 per client). In both setups, the server targeted one

client and metrics are based on this target client's dataset. Experiments were conducted on an Nvidia V100 16GB GPU, with training data limited to 1800 and 2640 images due to memory constraints.

**MosMedData (3D Attack):** [14] includes 1110 anonymized CT lung scans with dimensions of 512 x 512 x (36-41), comprising 42% male and 56% female participants, with the remaining unknown. The scans were categorized into five classes of COVID-19 severity. For the experiment, we selected volumes with 40 slices.

**Technical Setup:** We conducted federated learning simulations with 6 and 12 clients for both 2D and 3D models, with one client also acting as the central server. These settings reflect realistic client numbers typical in medical imaging collaborations. Each client received a local private dataset, generated by a random uniform split of the main datasets. Experiments, focusing on analytic attacks rather than optimization-based ones, were implemented using PyTorch [17].

**Reconstruction Attack: 2D vs Proposed 3D:** In the 2D Attack, the central server provides algorithms to participating clients with initial weights. For the 4-class classification problem, we employed the ResNet18[9] architecture. Subsequently, the central server possesses the ability to engage with any client during any epoch round. The 2D attack perform effectively when individual client weight updates are observable. The 2D method fails for 3D data due to its inability to maintain spatial correlations across all three dimensions, leading to suboptimal reconstructions. The original 3D file undergoes conversion into slices before being fed into the model by the model preprocessor, which is initialized by the malicious server. This preprocessing step can be a lightweight snippet initialized by the server. Subsequently, the server initializes malicious weights to target the client data with the malicious model. Once the slices are reconstructed by the server from received gradients, they are converted into a numpy array and patched together to generate the 3D data.

## 4   Results and Discussion

For 2D reconstruction metrics, we used Mean Squared Error (MSE) and Structural Similarity Index (SSIM). For 3D reconstruction, we chose SSIM and Peak Signal-to-Noise Ratio (PSNR). PSNR was preferred over MSE for its ability to consider the dynamic range of pixel values in 3D metrics. PSNR assesses image quality by calculating MSE between original and reconstructed images, offering pixel-level accuracy. SSIM evaluates perceptual quality by comparing luminance, contrast, and structure. MSE and PSNR quantify numerical differences, providing accuracy insights, while SSIM measures perceptual similarity, reflecting human visual perception. Together, these metrics offer a comprehensive evaluation of reconstruction quality.

**2D Attack** The 2D attack experiment (Table 1) reveals the critical influence of batch size on reconstruction attack efficacy. With six federated clients, increasing batch size decreases the number of reconstructions and degrades qual-

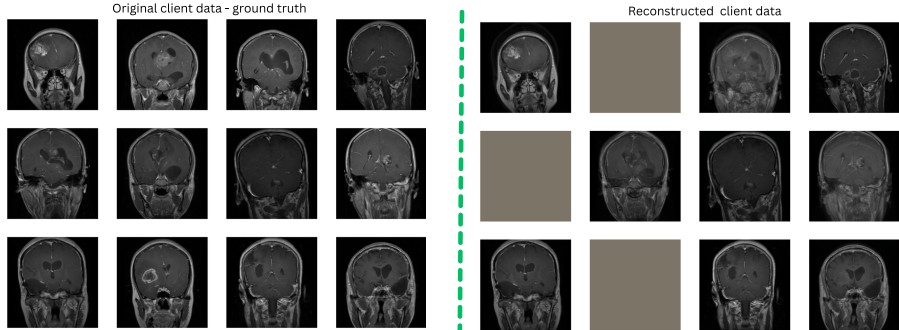

**Fig. 2.** 2D Brain Tumor MRI Data Reconstruction attack: The left side of the green line depicts the original client data, while the right side displays the corresponding reconstructed images derived from shared gradients of clients reconstructed by a malicious central server.The brown block indicates images not reconstructed, with reconstruction success dependent on the impact of malicious weights on neurons.

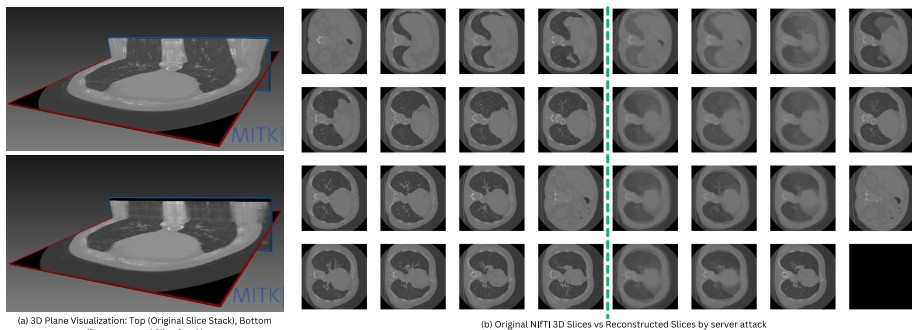

**Fig. 3.** 3D Attack: (a) depicts the original and reconstructed volume with two image planes cut through each of them, respectively. The axial plane (red frame) shows one slice of the slice stack (completely shown in (b)). The coronal plane (blue frame) shows a cut through all slices of the respective volume, visualized using MITK [19] Figure (b) contrasts original 3D CT scan slices on the left with their reconstructed counterparts on the right, processed by a malicious server from gradients, where black indicates failed reconstruction.

ity, evidenced by higher MSE and lower SSIM. Larger batch sizes lead to fewer affected data points per batch, as malicious weights deactivate other data points, allowing only positive neurons to activate and potentially leak a single data point. This complicates reconstructing gradients of multiple activated neurons, as shown in Figure 1b of the 2D scenario. Conversely, with twelve federated clients, larger batch sizes improve reconstruction quality, marked by lower MSE

and higher SSIM when compared with 6 clients. This suggests that while the quantity of data points susceptible to attack may decrease, the quality of the leaked data points remains high even at larger batch sizes, such as 128.

| FL Clients | Batch | Reconstructed images | MSE | SSIM |
|---|---|---|---|---|
| 6 | 8 | 114/300 | 0.0003 | 0.7171 |
| | 12 | 102/300 | 0.0007 | 0.5665 |
| | 16 | 102/300 | 0.0009 | 0.5822 |
| | 24 | 99/300 | 0.0012 | 0.5212 |
| | 32 | 107/300 | 0.0025 | 0.6171 |
| | 64 | 113/300 | 0.0040 | 0.4134 |
| | 128 | 102/300 | 0.0070 | 0.5139 |
| 12 | 8 | 89/220 | 0.0001 | 0.8666 |
| | 12 | 94/220 | 0.0001 | 0.8497 |
| | 16 | 86/220 | 0.0002 | 0.8240 |
| | 24 | 86/220 | 0.0002 | 0.8489 |
| | 32 | 82/220 | 0.0002 | 0.8189 |
| | 64 | 72/220 | 0.0002 | 0.8379 |
| | 128 | 64/220 | 0.0002 | 0.8070 |

**Table 1.** Comparative analysis of reconstruction metrics for 2D brain tumor data in federated learning experiments, focusing on scenarios with 6 and 12 federated clients. The evaluation encompasses varying batch sizes and the number of data points reconstructed from targeted client, utilizing 300 training data points from the targeted client in the 6-client FL scenario and 220 in the 12-client FL scenario.

**3D Attack** The results obtained from our investigation into the reconstruction metrics of 3D chest CT scans 3 present valuable insights into the vulnerability of sensitive data to malicious reconstruction attempts. Table 2 presents the results of one target client's average reconstructed slices from a single 3D volume with a depth of 40 slices, showing the average slices/depth that the server can reconstruct across different batches of a single 3D volume from the target client. Our experiments, conducted with six federated clients, involved varying batch sizes to evaluate the reconstruction process. The results demonstrate a clear trend wherein the quality of reconstruction diminishes as the batch size increases. Specifically, as the number of reconstructed slices per batch rises, both SSIM and PSNR exhibit a gradual decline. For instance, with a batch size of 8, the SSIM is measured at 0.8095 and the PSNR at 65.95, whereas with a batch size of 64, these metrics decrease to 0.6221 and 53.66, respectively. Moreover, the total number of reconstructed slices also plays a crucial role in determining the efficacy of the reconstruction process. We also show that a decrease in the total number of reconstructed slices corresponds to a reduction in both SSIM and PSNR metrics. This suggests that the reconstruction rate is less effective when the batch size increases.

| Batch | Average reconstructed slices | SSIM | PSNR |
|-------|------------------------------|--------|-------|
| 8 | 34/40 | 0.8095 | 65.95 |
| 12 | 32/40 | 0.7730 | 69.86 |
| 16 | 30/40 | 0.7544 | 65.30 |
| 24 | 22/40 | 0.7143 | 65.11 |
| 32 | 19/40 | 0.7055 | 51.23 |
| 64 | 14/40 | 0.6221 | 53.66 |

**Table 2.** Reconstruction metrics for 3D chest CT imaging in federated learning (reconstructed from the targeted client) were evaluated with 12 federated clients and varied batches, averaged over 3 runs in each case. Images consisted of 40 slices. Among these, some slices are reconstructed, and their metrics are detailed. These slices are then assembled to recreate the original 3D image.

## 5   Limitations and Mitigation Strategies

In this study, we explore the threat of malicious central servers in FL consortia, even when client security is high. This risk is crucial in FL collaborations across large jurisdictions or competitive environments. Client security alone does not ensure data safety, especially in cross-organizational FL where servers may be malicious. A malicious server can easily attack clients if it provides the training algorithm, which is common in FL. While we tested with ResNet18, this analytic attack can extend to other models. Despite the simplicity of our attack, real-world attackers would likely use more sophisticated methods, necessitating robust countermeasures by institutions.

To address risks associated with malicious server attacks we outline a multifaceted, effective strategy. This includes monitoring server models with a robust scanning architecture to detect unauthorized changes, using encrypted weights, auditing Fully Connected layers and ensuring code integrity through Hashsum Verification. A designated Clearance Officer, or a designated board, can ensure model update and result integrity by vetting changes to preserve trust in the learning process, while a dedicated person/group/third party algorithm monitors FL logs for significant weight changes during iterations. Increasing batch sizes and the number of FL clients in a 3D environment appears to enhance data protection effectively. Although it remains to be seen how common such malicious server attacks are in practice, they represent a significant threat capable of leaking high-fidelity data, particularly in sensitive domains like medical imaging. Due to the complex structure and spatial information, reconstruction is challenging when the algorithm reads the 3D scan as it is. Our future research will focus on reading 3D data directly and reconstructing the entire volume.

In conclusion, reconstructing over 35% of private high-fidelity 2D MRI and 3D CT data reveals significant privacy risks. Hospitals in federated learning should seek help to ensure patient privacy beyond software and network security..

**Acknowledgement.** This study was funded by X (grant number Y).

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
