# OpenReview forum: "Client Security Alone Fails in Federated Learning: 2D and 3D Attack Insights"
_MICCAI.org/2024/Workshop/MSB — MICCAI Student Board EMERGE Workshop 2024 Oral_

### Official Review · Reviewer_5HQi · 2024-07-07

**Recommendation:** 4
**Confidence:** 3

**Clarity:**

The paper is clear and well-written, with minor areas for improvement in clarity

**Feedback:**

More experiments to analyze the method and how good it is compared to existing methods would be useful.

**Justification:**

Interesting idea and clear structure, meets workshop publication standard.

**Reproducibility:**

Sufficient amount of details available for reproducing the main results, and open access is provided (or promised upon acceptance) to source code and/or data

**Strengths:**

1. paper is well-written, structure is clear, easy to follow
2. the topic of attacks in federate learning is interesting
3. the method can be applied to both 2D and 3D images
4. malicious server attacks is shown then effective solutions are provided
5. code is given, which makes this work easily followed

**Summary:**

This paper introduces precise reconstruction attacks on private data of FL clients, and reveals significant privacy breaches. They further propose to solve this problem by securing gradient, analytic, and linear layers.

**Weaknesses:**

No baseline methods are compared. No ablation studies are performed.

---

> ### Author Response · Authors · 2024-07-12
> **Rebuttal by Authors**
>
> >No baseline methods are compared. No ablation studies are performed. More experiments to analyze the method and how good it is compared to existing methods would be useful. The paper is clear and well-written, with minor areas for improvement in clarity.
>
> Dear Reviewer,
>
> Thank you very much for your thorough review and valuable feedback.
> We sincerely appreciate your recognition of our work as interesting and well-written. Regarding the absence of a baseline method, we would like to clarify that we are not introducing a new method but rather applying an existing method to medical data and adapting it to 3D data with our own preprocessing approach. Therefore, there is no baseline for this type of 3D attack.
> Additionally, we have updated the figures, revised the text, improved sentence structures, and corrected grammatical errors and typos. Once again, thank you for your positive review.

---

### Official Review · Reviewer_KJZ8 · 2024-07-09

**Recommendation:** 4
**Confidence:** 4

**Clarity:**

The paper is generally clear but has some clarity issues that could be addressed with moderate revision

**Feedback:**

Mainly the things to improve the paper are discussed in the weaknesses section. Please revise the Figures to make them readable and space saving. I also found some minor typos, this should be corrected.

**Justification:**

This paper introduces novel contributions through the application and extension of an existing method. The approach was evaluated on a suitable dataset. Although the authors acknowledge that this is a relatively simple method with certain limitations and may not be applicable to real-world scenarios, I believe this paper is valuable. It highlights the significance of the field and makes progress in the new domain of 3D data reconstruction.

**Reproducibility:**

Some amount of details available for reproducing the main results, and open access details are unclear

**Strengths:**

This paper addresses a critical issue in federated learning: data privacy. Its novelty lies primarily in applying reconstruction attacks to 3D data, a particularly intriguing case in medical imaging, which is a rather unexplored field.

A common analytical method is extended to work with 3D data. The malicious server modifies the algorithm to slice 3D data into 2D images and then reconstructs the data from these slices. After this preprocessing a common 2d reconstruction method can be applied.

The methodology section is well-explained and follows a clear structure. To evaluate the method, two datasets are used: one for 2D data and one for 3D data. The results are clearly presented.

I appreciate the evaluation of the impact of different numbers of training samples since this is a very sensitive aspect when dealing with reconstruction attacks.

**Summary:**

This paper examines the privacy-preserving capabilities of federated learning and investigates whether a malicious server can reconstruct the data.  Common reconstruction attacks are adapted to target 3D CT data. In this scenario, the malicious server supplies the training algorithm and introduces a malicious preprocessing step and a malicious layer to reconstruct the data.

**Weaknesses:**

This approach is limited to specific scenarios where the attacker can modify the algorithm (a curious-but-honest server would not be able to execute these attacks).

Figure 1 needs to be revised. The lettering is difficult to read, and the figure does not conserve space effectively. Similarly, the lettering in Figure 3 is not legible.

I have some remaining questions:

"[...]with one client also acting as the central server." Why is a client acting as a central server?

What kind of updates do you use? In the section *Reconstruction Attack: 2D vs Proposed 3D*, you mention weight updates and gradients received by the server. Please clarify.

You stated that you use a ResNet18 architecture for the 4-class classification problem in the 2D case. What architecture do you use for the 3D case?

"Despite the simplicity of our attack, real-world attackers would likely use more sophisticated methods, necessitating robust countermeasures by institutions." Given this, why is this work valuable?

---

> ### Author Response · Authors · 2024-07-12
> **Rebuttal by Authors**
>
> >This approach is limited to specific scenarios where the attacker can modify the algorithm (a curious-but-honest server would not be able to execute these attacks).
> Figure 1 needs to be revised. The lettering is difficult to read, and the figure does not conserve space effectively. Similarly, the lettering in Figure 3 is not legible.
>
> Thank you for your feedback and for recognizing the novelty of our idea. We have carefully reviewed your comments and made the following adjustments: we have adjusted the lettering and spacing in Figure 1 to enhance its readability and overall presentation.Yes, this approach is limited to scenarios where the server is malicious. Our aim is to showcase the risk, especially in cross-national and cross-industry collaborations. This risk is crucial in federated learning collaborations across large jurisdictions or competitive environments. Understanding how far data leaks are possible in both 2D and 3D medical imaging when the server is malicious.
>
> >with one client also acting as the central server." Why is a client acting as a central server? What kind of updates do you use? In the section Reconstruction Attack: 2D vs Proposed 3D, you mention weight updates and gradients received by the server. Please clarify.
>
> Thank you very much for your thorough review and valuable feedback. We would like to clarify our approach. In our simulation using OpenStack and code, we designated one client to act as the server, excluding clients 6 and 12. This setup is designed to mimic a real-world scenario where a client assumes the role of a server, facilitating the simulation of central server functionalities within our experimental framework. We understand this might be confusing, but it is not crucial to understanding the paper. The results would be the same if the server were separate. In response to your question about the types of updates used, we would like to clarify the confusion. We have utilized gradient updates in our experiments. The statement about weight updates was referring to the 2D attack in the reference paper by Boenisch et al. (2023) [2], which used a malicious weight-based approach.
>
> >You stated that you use a ResNet18 architecture for the 4-class classification problem in the 2D case. What architecture do you use for the 3D case?
>
> We have employed ResNet-18. We have clarified this in the manuscript to ensure a better understanding of our approach because the model is still 2D after converting 3D volumes to slices.
>
> >"Despite the simplicity of our attack, real-world attackers would likely use more sophisticated methods, necessitating robust countermeasures by institutions." Given this, why is this work valuable?
>
> Regarding your question about the value of this work, while we have provided a version of the attack, real-world attackers can further over-complicate and obfuscate it through layers of abstraction, false justifications, complicated architectures, or hiding it in libraries. Therefore, it’s not simply a matter of adding a piece of code to check for this attack. People need to be vigilant about such attacks when tasked with data protection and must deeply understand the code that is running to effectively safeguard against these threats.
>
> >Mainly the things to improve the paper are discussed in the weaknesses section. Please revise the Figures to make them readable and space saving. I also found some minor typos, this should be corrected#
>
> Dear Reviewer,
> Thank you very much for your thorough review and valuable feedback.
> We really appreciate your constructive review. We have addressed all the weaknesses and feedback to the best of our ability. The figures have been updated for better readability, and we have corrected the typos and grammatical mistakes.

---

### Official Review · Reviewer_ruT4 · 2024-07-10

**Clarity:** The paper is exceptionally well-writt…
**Recommendation:** 5
**Confidence:** 3

**Feedback:**

Figure 1 is rather hard to understand. Some more labels would help, especially in Fig 1 b.

**Justification:**

The contribution of this paper is important and interesting for the workshop. Especially the study and derived discussion.

**Reproducibility:**

Sufficient amount of details available for reproducing the main results, and open access is provided (or promised upon acceptance) to source code and/or data

**Strengths:**

- Paper is well-written and well-structured.
- Related work is outlined well and does a good job at contextualizing this work.
- This is a first practical demonstration of FL attacks in medical imaging on 3D data (CT / MRI), which makes an important point about patient privacy and data security in a clinical FL setting. Hence, this is an important contribution to the field and to the workshop.
- The number of participating clients is sufficient and realistic for a clinical FL setting.
- The attack scenario is realistic and well-motivated.
- The authors clearly name the limitations of their study and propose ways to mitigate server attacks, e.g. by increasing clients and batch size, which is motivated by the results presented in Tables 1 and 2. All in all, the paper is complete in its argumentation.

**Summary:**

The main contribution of this paper is an experimental framework describing reconstruction attacks in a federated learning setting. The attack scenario investigated is a malicious server, in which the authors achieve a significant success in reconstructing 3D images from gradient updates.

**Weaknesses:**

- To my understanding, the main contribution is the adaptation of a 2D method to a 3D image data setting. One could argue that the contribution is only additive. However, this should be considered only a minor limitation -- the actual execution of the attack in a multi-client study and the discussion (+mitigation strategy) are valuable contributions to the research field and this workshop.

- Only one architecture was investigated (ResNet-18). However, I understand that training and evaluating multiple architectures would cause significant overhead in such a federated setup; and if the method works with ResNet-18, I assume it is likely to work with larger architectures which will retain more features due to increased capacity. Further, this limitation was explicitly addressed in Limitations. All in all, the study setup is still quite complete for a medical imaging paper.

---

> ### Author Response · Authors · 2024-07-12
> **Rebuttal by Authors**
>
> > To my understanding, the main contribution is the adaptation of a 2D method to a 3D image data setting. One could argue that the contribution is only additive. However, this should be considered only a minor limitation -- the actual execution of the attack in a multi-client study and the discussion (+mitigation strategy) are valuable contributions to the research field and this workshop.
>
> Dear Reviewer,Thank you for your positive feedback and for recognizing the value of our work. We are pleased that you found our contribution meaningful. Yes, we adapted the 2D method to showcase the 3D reconstruction capability using a 2D attack. We appreciate your recognition of our multi-client study and mitigation strategy as valuable contributions. As a follow-up, we plan to use the 3D volume directly to reconstruct 3D data. Thank you again for your valuable review.
>
> >Only one architecture was investigated (ResNet-18). However, I understand that training and evaluating multiple architectures would cause significant overhead in such a federated setup; and if the method works with ResNet-18, I assume it is likely to work with larger architectures which will retain more features due to increased capacity. Further, this limitation was explicitly addressed in Limitations. All in all, the study setup is still quite complete for a medical imaging paper.
>
> Thank you for your feedback. Yes, due to the significant overhead in a federated learning setting, we restricted our investigation to one architecture, ResNet-18. However, since the server is malicious, this attack can be applied to any architecture. We appreciate your understanding and acknowledgment of our study setup.
>
> > Figure 1 is rather hard to understand. Some more labels would help, especially in Fig 1 b.
>
> In response to your suggestion, we have added labels to Figure 1b to enhance its clarity and make it more understandable.

---

### Meta-Review · Area_Chair_Q74S · 2024-07-16

**Recommendation:** Accept (Oral)
**Confidence:** 4

**Metareview:**

This paper investigates 3D reconstruction attacks in federated medical imaging, a new area with significant privacy implications. The reviewers agree the paper is well-structured, clearly explained, and uses realistic scenarios. The paper goes beyond just identifying vulnerabilities; it explores mitigation strategies for these attacks. The reviewers overall agree that this paper presents a valuable contribution to the field of federated learning in medical imaging. Consider incorporating suggestions for additional experiments if feasible, to further strengthen the analysis. The paper uses a 2D approach and authors may explore the value of applying it to a 3D medical setting with a multi-client study. The authors' have satisfactory explanations for limitations, however, it must be spelt out in the paper. As promised, the authors must address the presentation issues for better readability and clarity.

---

### Decision · Program_Chairs · 2024-07-16

Accept (Oral)